# Comparative Analysis of the Mitochondrial Genomes of Three Species of *Yangiella* (Hemiptera: Aradidae) and the Phylogenetic Implications of Aradidae

**DOI:** 10.3390/insects15070533

**Published:** 2024-07-14

**Authors:** Liangpeng Ji, Zhancheng Jia, Xiaoshuan Bai

**Affiliations:** College of Life Sciences and Technology, Inner Mongolia Normal University, Hohhot 010022, China; jliangpeng@outlook.com (L.J.); jiazhancheng@outlook.com (Z.J.)

**Keywords:** Aradidae, mitochondrial genome, gene rearrangement, phylogenetic analysis, comparative analyses

## Abstract

**Simple Summary:**

Aradidae is a large family in Hemiptera, which feeds on mycelium, and its phylogenetic status is not fully understood. Mitochondrial genome sequences can be used to study species identification and phylogeny, and provide valuable molecular markers for further genetic research. In this paper, we sequenced three species of the genus *Yangiella* for the first time, assembled three mitochondrial genomes, and compared the general characteristics of the mitochondrial genomes of the three species. It was found that the base composition and mitochondrial genome structure of the three species were highly similar. The phylogeny of *Yangiella* was also discussed. Based on the phylogenetic analyses results of two data matrices, the status of *Yangiella* was discussed and the monophyly of this genus was verified. In addition, we infer that the genus *Yangiella* diverged about 57 million years ago.

**Abstract:**

The mitochondrial genomes of three species of *Yangiella* were sequenced, annotated, and analyzed. The genome length of the three species of the genus is 15,070–15,202 bp, with a typical gene number, including a control region, 2 ribosomal RNA genes (rRNAs), 22 transfer RNA genes (tRNAs), and 13 protein-coding genes (PCGs). It was found that the mitochondrial genome of *Yangiella* had AT bias. Except for the lack of a DHU arm of the *trn*S1 gene, the other tRNAs had a typical cloverleaf structure, and the codon usage preferences of the three species exhibited high similarity. In addition, tRNA gene rearrangements were observed among the three subfamilies of Aradidae (Mezirinae, Calisiinae, Aradinae), and it was found that codon usage preferences appeared to be less affected by base mutation and more by natural selection. The Pi and Ka/Ks values indicated that *cox*1 was the most conserved gene in the mitochondrial genome of Aradidae, while *atp*8 and *nad*6 were rapidly evolved genes. Substitution saturation level analysis showed that the nucleic acid sequence of mitochondrial protein-coding genes in Aradidae did not reach saturation, suggesting the rationality of the phylogenetic analysis data. Bayesian and maximum likelihood methods were used to analyze the phylogeny of 16 species of Hemiptera insects, which supported the monophyly of Aneurinae, Carventinae, and Mezirinae, as well as the monophyly of *Yangiella*. Based on fossils and previous studies, the differentiation time was inferred, indicating that *Yangiella* diverged about 57 million years ago.

## 1. Introduction

Aradidae is a larger family in Hemiptera, with 8 subfamilies and about 2000 species [1]. Mezirinae is the largest subfamily and is found in all geographical regions of the world [2]. *Yangiella* is a genus of Mezirinae [3]. This genus of insects usually lives under the bark of fallen trees, and is sometimes found in gaps on the bark surface [4]. They live in groups and feed on hyphae, mainly distributed in Southwestern China and parts of Southeast Asia [5]. Due to its high concealment and lesser contact with humans, past research on this genus has mainly focused on traditional morphological classification, and molecular biology research has not been carried out, resulting in its classification status still being unclear.

Mitochondrial genomes have the characteristics of maternal inheritance, fast evolution, easy separation and purification, and are widely used in the study of phylogenetic relationship of insects [6,7] and molecular evolution [8,9]. The typical insect mitochondrial genome is a double-stranded, circular, closed DNA molecular structure with a length of 15–18 kb, containing 37 genes (composed of 13 protein-coding genes and 24 RNA genes) [10], and a non-coding control region containing the origin of replication [11]. In the past, most of the studies on the molecular system of Aradidae were based on partial gene fragments, and Heisset et al. analyzed the evolution of three subfamilies based on *cox*1 fragments [12,13]. Marchal et al. used cox1, 16S rRNA, and 28S rRNA gene fragments to analyze the phylogeny of 79 species in eight subfamilies. They believed that Prosympiestinae and Chinamyersiinae were paraphyletic, and the remaining six subfamilies were monophyletic [14]. However, compared with the complete mitochondrial genome, a single gene can only reflect part of the information and provide limited information. The complete mitochondrial genome sequence contains more abundant information [15], which is a powerful evidence form for phylogenetic research [16].

In this study, the mitochondrial genomes of three *Yangiella* species were sequenced and annotated for the first time. The basic characteristics of the mitochondrial genome, codon usage bias, and tRNA gene secondary structure were described and analyzed. The internal relationship of Aradidae was elucidated by analyzing the evolutionary rate and genome rearrangement of each species of Aradidae. In addition, based on 13 protein-coding genes and 2 RNA genes, the phylogenetic trees of 16 species (including 14 Aradidae insects and 2 outgroups) were reconstructed by Bayesian inference (BI) and maximum likelihood (ML) methods, and the divergence time of Aradidae was estimated based on phylogenetic trees and fossil data.

## 2. Materials and Methods

### 2.1. Specimen Collection, Extraction, and Sequencing

Samples of *Yangiella mimetica* (Hsiao, 1964 [3]), *Yangiella montana* (Zhang, Bai, Heiss, and Cai, 2010 [5]) and *Yangiella* sp. were collected in China (Table 1). The specimens were immersed in 95% ethanol during collection, and then transferred of −20 °C for long-term preservation in the Inner Mongolia Normal Entomological Museum (Hohhot, Inner Mongolia). The total DNA was extracted from the leg and chest muscle tissues of insects. Using the kit of Tiangen Biochemical Technology Co., Ltd. (Beijing, China), DNA was extracted according to the instructions in the instructions. The processed samples were sent to Berry Genomics Co., Ltd. (Beijing, China) for sequencing (DNA concentration ≥ 20 ng/μL, total ≥ 500 ng). Second-generation sequencing technology was used, and the sequencing mode was Novaseq 6000-S4-150 PE.

### 2.2. Gene Assembly, Annotation, and Analysis

Raw read data with a double-end sequencing length of 150 bp were obtained, and Fastp v0.23.0 was used to perform quality control on the raw data [17], and the reads below the average quality value of the reads were removed. The reads with N content less than 5 were screened to obtain clean data with a size of 5G. SPAdes v3.15.5 (https://github.com/ablab/spades (accessed on 15 March 2024)) was used for assembly [18]. Based on the assembly process, the contig sequence was obtained by iterative method to construct the local blast database. Using the *cox*1 sequence of *Neuroroctenus yunnanensis* (Hsiao, 1964), which belongs to Mezirinae, as bait, the BLAST function was used to screen contigs of the target mitochondrial genome.

The mitochondrial genome sequence was imported into the online tool MITOS (http://mitos.bioinf.uni-leipzig.de/index.py (accessed on 22 March 2024)), and set to invertebrate mitochondrial coding mode for automated annotation [19]. The annotation results were imported into Geneous Prime v11 [20], and the gene boundaries were manually corrected by comparing them with related species. The location of the tRNA gene was reconfirmed using the tRNAscan-SE online tool (Cover cutoff = 5) [21], and Adobe Illustrator CS 2024 software was used for drawing. The CGView Server online server (https://cgview.ca/ (accessed on 10 April 2024)) was used to visualize the mitochondrial map [22]. The gene rearrangement of Aradidae was studied by comparing with the mitochondrial gene arrangement of *Macroscytus subaeneus* (Dallas, 1851) from the Cydnidae. Finally, the newly sequenced mitochondrial genomes were submitted to the GenBank database.

The nucleotide composition of the gene was calculated using MEGA11 [23]. AT-skew and GC-skew are calculated by AT-skew = (A − T)/(A + T) and GC-skew = (G − C)/(G + C) [24], respectively. DnaSP v6.12.03 [25] was used to calculate the non-synonymous substitution rate (Ka), synonymous substitution rate (Ks), and nucleotide diversity (Pi) of protein-coding genes. Relative synonymous codon usage (RSCU) and effective number of codon (ENC) were analyzed by CodonWv 1.4.4 software.

Neutrality plot analysis was performed on Aradidae species. GC12 (the mean value of G + C content at codons 1 and 2 of protein-coding genes) was used as the ordinate, and GC3 (the G + C content at codon 3 of protein-coding genes) was used as the abscissa. The difference in base content at different sites of protein-coding genes was judged by a slope to determine whether it demonstrated neutral evolution [26]. ENC-plot analysis was performed, and the scatter plot was drawn using Microsoft Excel 2019 with GC3 as the abscissa and ENC as the ordinate [27]. The expected value was calculated according to the formula ENC = 2 + GC3 + (29/(GC3^2^ + (1 − GC3)^2^)) [28], and the standard curve was drawn. Finally, DAMBE v7.3.32 software was used to evaluate the replacement saturation of sequence data [29].

### 2.3. Phylogenetic Analysis

We retrieved the complete mitochondrial genomes of 14 insects from NCBI. Combined with the newly determined complete mitochondrial genomes of three species of the genus *Yangiella* in this study, an initial dataset containing 16 species (14 species of Aradidae, *Macroscytus subaeneus*, *Urochela quadrinotata* (Reuter 1881) as outgroup, see Appendix A) was complete.

The initial dataset was imported into Phylosuite v1.2.3 software [30] to extract the nucleotide sequences of protein-coding genes (PCGs) and rRNAs. Subsequently, multiple sequence alignments of each gene of the PCG and rRNA sequences were performed using the G-INS-i strategy in MAFFT v7.464 software [31]. As MAFFT v7.464 does not consider the codon structure of PCGs, it may introduce alignment errors; therefore, it was necessary to optimize the aligned PCG sequences using MACSE [32]. Then, GBlocks v0.91 b was used to prune PCGs to remove sites with alignment errors or multiple substitutions, so as to remove phylogenetic noise and retain phylogenetic signals [33]. rRNA sequences were trimmed using trimAl v1.4 software [34]. Finally, FASconCAT-G v1.04 was used to concatenate the sequences [35]. Two datasets for phylogeny were obtained: (1) a data matrix of 13 protein-coding genes and 2 rRNA genes (PCGsRNA); (2) a data matrix (PCGs12RNA) constructed by the first and second codons of 13 protein-coding genes and 2 rRNA genes.

The overall heterogeneity of the two data matrices was evaluated using Aligroove v1.06 [36]. Then, ModelFinder [37] was used to evaluate the optimal partitioning strategy and evolutionary model of the PCGsRNA and PCGs12RNA matrices, respectively. Based on the results of ModelFinder (Appendix A), a Bayesian inference (BI) phylogenetic tree was constructed using Exabayes v1.5.1 software [38]. Outgroups were manually selected, MCMC generations were set to 10,000,000, sampling frequency was 1000, four MCMC chains were run, and the Burnin Fraction value was 0.25. When the average standard deviation of the split frequencies (ASDSF) value was less than 0.01, BI ran converge and the BI phylogenetic tree was obtained. Then, IQ-tree v2 software [39] was used to construct the maximum likelihood (ML) phylogenetic tree, manually select the outgroup, and calculate the Ultra-fast Bootstrap Value as the support rate for each node [40]. The online website TVBOT (https://www.chiplot.online/tvbot.html (accessed on 10 May 2024)) was used to visualize the phylogenetic trees of BI and ML [41].

### 2.4. Divergence Time Estimation

For these two matrices, this study used BEAST v1.10.4 to calculate the divergence time of each genus of Aradidae [42]. The partitioning scheme was set according to the results of ModelFinder. The divergence time of the previous study was used as the calibration point [43,44] (Appendix A). The prior distribution selected was the normal distribution and the relaxed molecular clock model. The total generation value of the MCMC chain was set to 20 million generations, and sampling was performed every 1000 generations. The running results were viewed using Tracer v1.7.2 to test whether the parameters converged (ESS > 200) [45], TreeAnnotator v1.10.4 was used to obtain the MCC tree (the maximum clade credibility tree), and the proportion of discarded samples was set to the top 25%. Finally, FigTree v1.4.4 was used to view the MCC tree.

## 3. Results

### 3.1. General Characteristics of the Mitochondrial Genome of the Genus Yangiella

In this study, we sequenced the complete mitochondrial genomes of three species of the genus *Yangiella* (Figure 1), with lengths of 15,192 bp for *Y. mimetica*, 15,205 bp for *Y. montana*, and 15,070 bp for *Yangiella* sp., respectively. The difference in mitochondrial genome length is mainly due to changes in the length of the control region and rRNA. The mitochondrial genomes of the three species of *Yangiella* contain 37 genes and a non-coding control region (CR). Among them, there are 23 J-strand genes, including 14 tRNA genes and 9 PCGs. There are 14 genes on the N-strand, including 4 PCGs, 8 tRNA genes, and 2 rRNA genes (Appendix A). Compared with the mitochondrial gene arrangement of Cydnidae, tRNA gene rearrangement was found in Aradidae, forming three-gene arrangement. Among them, *Aradacanthia heissi* (Bai, Zhang, and Cai) and *Aradus compar* (Kiritshenko, 1913) each all a gene arrangement, and Aneurinae, Carventinae, and Mezirinae all have the same gene order (Figure 2).

There are 12–14 gene overlap regions in the complete mitochondrial genomes of the three species, ranging from 1 to 8 bp in length. The longest overlap region is located between *trn*W and *trn*C in the three species, which is 8 bp. There are also between seven and eight intergenic spacers (IGS) between each gene, with a length of 1–50 bp. The largest intergenic spacers of each species are located between *trn*Q and *trn*I, with a length of 44–50 bp. In addition, among the 13 protein-coding genes of each *Yangiella* species, except for *cox*1, *cox*2, and *nad*1, which used TTG as the starting codon, the remaining PCGs used ATN (N = A, T, C, G) as the starting codon. The starting codons of *nad*3, *nad*4, *nad*4L, and *nad*6 were different among different species. Ten PCGs of the three species were terminated with TAA or TAG, and *cox*2, *nad*4, and *nad*5 were terminated with T residues (Appendix A).

The mitochondrial genome of *Yangiella* had AT bias, and the AT content was between 69.1 and 69.7%, showing positive AT and negative GC bias. The AT and GC bias in protein-coding genes were both negative, while the genes on the J-chain had positive AT and negative GC tilt, and the N-chain was the opposite (Appendix A).

The mitochondrial genome of *Yangiella* contains 22 tRNAs, with a length of 61–70 bp, of which *trn*A is the shortest, with a length of 61 bp in all three species. *Y. mimetica* has the longest *trn*W (70 bp). The longest *trn*Q in *Y. montana* was 69 bp; the *trn*Q, *trn*K, and *trn*D genes of the new species were all 69 bp. The *trn*S1 in each species lacks the DHU arm (Dihydrouridine arm) and cannot form a typical cloverleaf secondary structure. Other tRNAs can fold to form a typical clover structure (Figure 3, Appendix A). A total of 102 base mismatches occurred in the tRNA of the three species, including 75 GU mismatches, 10 UU mismatches, 8 AC mismatches, 5 UC mismatches, 3 AA mismatches, and 1 AG mismatch. GU, UU, UC, and AC mismatches were found in all three species, and the number of GU mismatches was the highest in all three species, while AG mismatches were only found in *trn*C of *Y. montana*.

The *Yangiella* mitochondrial genome contains two rRNA genes; *rrn*L is located between *trn*L1 and *trn*V, and *rrn*S is located between *trn*V and control region. The length of *rrn*L is between 1258 and 1384 bp, and the length of rrnS is between 743 and 748 bp, both of which have obvious AT bias (Appendix A).

### 3.2. Codon Preference Analysis

Among the 13 PCGs of *Y. mimetica* and *Y. montana*, UUA, UCA, CGA, UCU, and ACA were used more frequently, while UUA, UCA, ACA, AGA, and CGA were used more frequently in *Yangiella* sp. However, the synonymous codon usage frequency of UUA was the highest in the three species, while the relative synonymous usage frequency of AGG was the lowest in the three species. The most commonly used amino acids were Ile, followed by Met, Phe, and Leu2 (Figure 4). In general, the codon usage preferences of the three species of *Yangiella* are similar.

### 3.3. Neutrality Plot Analysis and ENC-Plot Analysis

The results of neutral mapping analysis (Figure 5A) showed that the GC12 values of the mitochondrial gene of Aradidae were between 0.309 and 0.391, and the GC3 values were between 0.148 and 0.341. The GC12 and GC3 were significantly correlated (R^2^ = 0.9131, *p* < 0.01), indicating that the mutation pressure affected all codon sites. In addition, all the mitochondrial genes of Aradidae were located below the ENC standard curve (Figure 5B).

### 3.4. Base Substitution Saturation Analysis

The transversion rate of protein-coding genes in Aradidae is often greater than the transition rate, which is more in line with the ideal state of substitution. At the saturation level, when the total genetic distance increases, the slope of the transversion saturation curve gradually decreases, but there is no obvious platform, so it does not reach saturation; the slope of the transition saturation curve is much larger than zero, and does not reach saturation (Figure 6A). In theory, the third site of the protein-coding gene is subjected to the least selection pressure and the substitution occurs more frequently. The substitution saturation of the three coding sites was calculated, respectively. The first and second sites had almost positive slope lines, which were completely unsaturated (Figure 6B,C). The slope of the transition trend line at the third site is much larger than zero, and it also does not reach the saturation state, but the slope of the transversion trend line gradually becomes smaller, showing a slight saturation state (Figure 6D). The trend line of substitution saturation of rRNA gene is almost a standard positive slope line, which is completely unsaturated (Figure 6E).

### 3.5. Nucleotide Diversity (Pi) and Nonsynonymous (Ka)/Synonymous (Ks) Mutation Rate Ratios

In this study, the Pi values of 13 PCGs in the mitochondrial genome of Aradidae were calculated (Figure 7). The results showed that the overall Pi changes of 13 PCGs in Aradidae were not significant. Among them, *atp*8, *nad*6, and *nad*4L had higher Pi values than other coding genes, and the value of *cox*1 was the smallest, indicating that the gene had the least variability. The Ka/Ks ratio was used to evaluate the evolutionary rate of 13 PCGs in Aradidae species (Figure 7). The results showed that the Ka/Ks ratio of all genes was less than 1, indicating that these genes were in a state of purification selection. However, *atp*8, *nad*2, *nad*41, *nad*5, and *nad*6 showed higher Ka/Ks ratios, and they may have higher evolutionary rates than other genes.

### 3.6. Heterogeneity Analysis

Heterogeneity in nucleotide divergence was evaluated via pairwise comparisons in a multiple sequence alignment. The results showed that the heterogeneity of the two matrices was low and could be used to construct phylogenetic trees (Figure 8). On the whole, PCGs12RNA has darker blue and lower heterogeneity. The heterogeneity rate of the third codon was high, and the removal of the third codon matrix can reduce the degree of heterogeneity, which can further explain that PCG base heterogeneity is mainly concentrated in the third codon.

### 3.7. Phylogenetic Analyses

In this study, we conducted a phylogenetic analysis that included 14 species of Aradidae, as well as outgroup species from Pentatomidae. We used maximum likelihood (ML) and Bayesian inference (BI) methods, and used two data matrices (PCGsRNA and PCGs12RNA) to generate four phylogenetic trees. The Bayesian inference (BI) and maximum likelihood (ML) trees of the two data matrices show exactly the same topology, and most nodes are strongly supported. The results showed that except for Calisiinae and Aradinae, where only one species was not discussed, the remaining subfamilies were Aneurinae (BP = 90; PP = 1), Carventinae (BP = 90; PP = 10), and Mezirinae (BP = 100; PP = 1), which are monophyletic subfamilies. Among them, Calisiinae is located at the base of the phylogenetic tree, and Mezirinae is a highly evolved subfamily, but the ML tree support of individual points in its internal branch is weak (BP = 16–19), and it is stronger in the BI tree (PP = 0.80–0.81). In addition, the three *Yangiella* sequenced in this study were monophyletic (BP = 100; PP = 1), and it is a sister group to Chinolyda (Figure 9).

### 3.8. Divergence Time Estimation

We estimated the divergence time, and there was no significant difference between the results of the two data matrices (Appendix A). Based on the results of the two data matrices, it is inferred that the most recent common ancestor of Aradidae appeared in the Late Jurassic (168 MYA). Aradinae was isolated from Carventinae, Mezirinae, and Aneurinae in the early Early Cretaceous (146 MYA). In the late Early Cretaceous (128 MYA), Carventinae was separated from Aneurinae and Mezirinae. Finally, *Yangiella* and Neuroctenus formed and differentiated in the early Eocene (57 MYA) (Figure 10).

## 4. Discussion

Gene rearrangement is considered to be an important molecular marker in revealing insect evolution [46]. The arrangement and direction of the mitochondrial genes of the three species of *Yangiella* were consistent with those of Aneurinae, Carventinae, and Mezirinae, but the *trn*C and *trn*Y genes were shifted compared with *Aradus comper* in Aradinae [43]. Compared with *Aradacanthia heissi* [47] of Calisiinae, *trn*C and *trn*W genes were reversed. The *trn*Q and *trn*I genes were rearranged in Aradidae compared to *Macroscytus subaeneus* [48], which is also a Hemiptera (Figure 2). In addition, it was also found that the gene rearrangements of Aradidae mainly occurred between tRNA genes and were located between the control region and *cox*1. Previous studies have noted that genes around the replication start point (such as CR) are more likely to be replicated to form a “hot spot” region, making the rearrangement of gene order more likely [49]. Moreover, the same gene arrangement (*trn*Q–*trn*I) was found in the mitochondrial genomes of all Aradidae, which was different from the gene order of most other hemipteran insects (*trn*I–*trn*Q) [50]. In previous studies, this rearrangement was considered to be an ancestral feature of the Aradidae [43].

The mitochondrial genome of *Yangiella* insects shows high AT bias, which is widely present in the mitochondrial genome of insects. This may be because guanine (G) and cytosine (C) nucleotides require more energy for biosynthesis than adenine (A) and thymine (T) nucleotides, and are not easily available in cells [51]. Therefore, the unequal use of four bases usually leads to high A + T content [8,52]. At the same time, the two strands (heavy chain and light chain) of the mitochondrial DNA of the genus insects are not uniform, and there is no reverse chain asymmetry. The AT skew is positive and the GC skew is negative, which is consistent with the characteristics of most insect genomes [53]. This phenomenon may be related to replication and transcription mechanisms [54].

The *trn*S1 genes of *Yangiella* species all lack the (DHU) arm and form a ring. Such a phenomenon exists in most insects [8]. At the same time, six mismatched base pairs were found, of which the largest number of GU mismatches can form hydrogen bonds in RNA, but the geometric shape of the base causes the GU pair to be weaker than the hydrogen bond of the AU pair [55,56]. GU base pairing also plays an important role in biological processes [57]. In addition, non-classical UC mismatches, UU mismatches, AC mismatches, AA mismatches, and AG mismatches were also found. These mismatches may be caused by error correction mismatches during RNA editing, but the mismatch phenomenon has little effect on the corresponding function of tRNA genes [58]. If the helical regions of RNA molecules are used to construct phylogeny in subsequent studies, it is necessary to understand the way these parts of the sequence evolve. The genetic relationship between species can be judged by codon usage patterns, that is, similar codon preference between genomes means a closer genetic relationship [59,60]. In this study, the RSCU values of the mitochondrial genomes of three species of the genus *Yangiella* were analyzed. The analysis revealed a high degree of similarity in codon preference among the three species, further confirming that they belong to the same genus.

The ENC value of Aradidae is low, indicating that its codon preference is strong [61]. All the genes in the results of ENC mapping analysis were located under the standard curve, indicating that the mitochondrial gene codons of Aradidae insects were greatly affected by natural selection. Neutrality plot analysis showed that the correlation between GC12 and GC3 in the mitochondrial genome of Aradidae was extremely significant, indicating that the codon usage preference of Aradidae was affected by base mutation. Combined with the regression coefficient, it can be inferred that natural selection and base mutation have a great influence, but the influence of natural selection is greater.

The Pi and Ka/Ks values of *cox*1 are the lowest, indicating that it is the most conserved gene in the mitochondrial genome [62], while *atp*8 and *nad*6 are rapidly evolving genes. In addition, the Ka/Ks < 1 of each gene in Aradidae indicates that the evolution of Aradidae is subject to purification selection [63,64], and the stability of gene function needs to be maintained by eliminating harmful mutations [65]. This is consistent with the results of the study on the evolutionary rate of Hemiptera insects in 2015 [66]. However, in the study of scale insects, nine PCGs had a higher nonsynonymous mutation rate (Ka/Ks > 1), and the dominant evolution seemed to be positive selection [50]. This difference may be due to the adaptive evolution of scale insects with host plants, while the flat bugs mostly live under the bark of rotten wood, without specific host plants, and the living environment is rarely changed.

The results of substitution saturation level analysis showed that the nucleic acid sequence of mitochondrial protein-coding genes in Aradidae did not reach saturation, which provided data rationality support for subsequent phylogenetic analysis. Heterogeneity analysis of the two data matrices showed that the compositional heterogeneity of Aradidae was low and would not lead to abnormal topological clustering. In addition, the phylogenetic results of this study are basically consistent with the traditional morphological classification and previous molecular studies [13]. Both datasets strongly support the monophyly of each subfamily and *Yangiella* of Aradidae. Through the analysis of divergence time, it was found that the differentiation time of the main lineages of Aradidae was consistent with its fossil record. For example, Aradinae branched with Carventinae, Mezirinae, and Aneurinae in the Early Cretaceous, which is consistent with the age of the oldest Aradinae fossil, *Aradus nicholasi* (Popov, 1989) (ca. 125–113 MYA) [67]. The branch of Mezirinae in the Late Cretaceous is consistent with the time of the Mezirinae fossil *Myanmezira longicornis* (Heiss and Poinar Jr, 2011) (100–90 MYA) in Myanmar [68]. The results of this study can provide a scientific basis for the evolution of Aradidae insects, and more comprehensive phylogenetic and more accurate molecular clocks need further study.

## 5. Conclusions

In this study, we report for the first time the complete mitochondrial genomes of three species in the genus *Yangiella*. It was found that the mitochondria of the genus have similar structural characteristics and nucleotide composition. The unique gene arrangement of Aradidae was also found, which further proved the monophyly of this family. All the phylogenetic trees have the same topological structure, which supports the monophyly of Aneurinae, Carventinae, and Mezirinae, and the monophyly of *Yangiella*. Unfortunately, the number of species used in this paper is limited and was not analyzed in conjunction with morphological characters. Therefore, further and more comprehensive studies are needed, as well as exploration of species divergence times.

## Figures and Tables

**Figure 1 insects-15-00533-f001:**
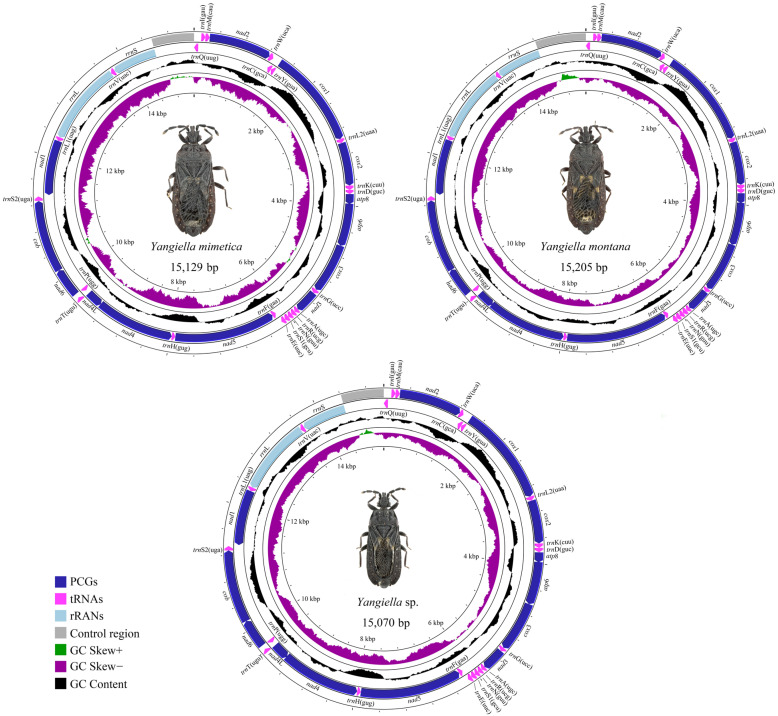
Mitochondrial genome maps of three species of *Yangiella*. The chain is marked with an arrow indicating the direction of gene transcription. Gene lengths correspond to nucleotide lengths in the diagram. The outermost layer is the J-chain, and the second layer is the N-chain.

**Figure 2 insects-15-00533-f002:**
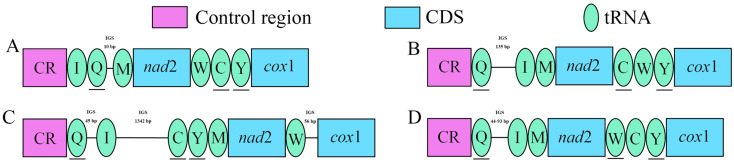
Mitochondrial gene rearrangements in flat bugs. Except for the underlined genes, the other genes were transcribed from left to right. IGS is the intergenic region. (**A**) *Macroscytus subaeneus*. (**B**) *Aradacanthia heissi*. (**C**) *Aradus compar*. (**D**) *Aneurus similis*, *Aneurus sublobatus*, *Libiocoris heissi*, *Taiwanaptera montana*, *Arbanatus* sp., *Brachyrhynchus hsiaoi*, *Brachyrhynchus triangulus*, *Mezira* sp., *Neuroctenus yunnanensis*, *Yangiella* sp., *Yangiella mimetica*, *Yangiella montana*.

**Figure 3 insects-15-00533-f003:**
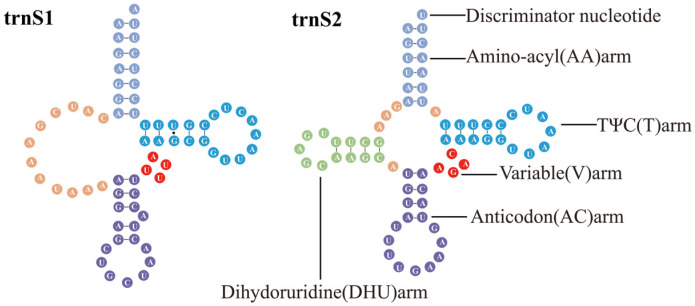
Secondary structure prediction of the tRNA gene of *Yangiella* sp. The Watson–Crick pair is represented by a straight line, and the mismatch is represented by a dot. All of the *Yangiella* sp. mitogenome tRNAs except for *trn*S1 follow the canonical conserved three-loop cloverleaf structure consisting of the amino-acyl arm (lilac), dihydrouridine arm (DHU, green), pseudouridine arm (TΨU loop, blue), and the anticodon loop (modena). All tRNAs also contain a variable region (red).

**Figure 4 insects-15-00533-f004:**
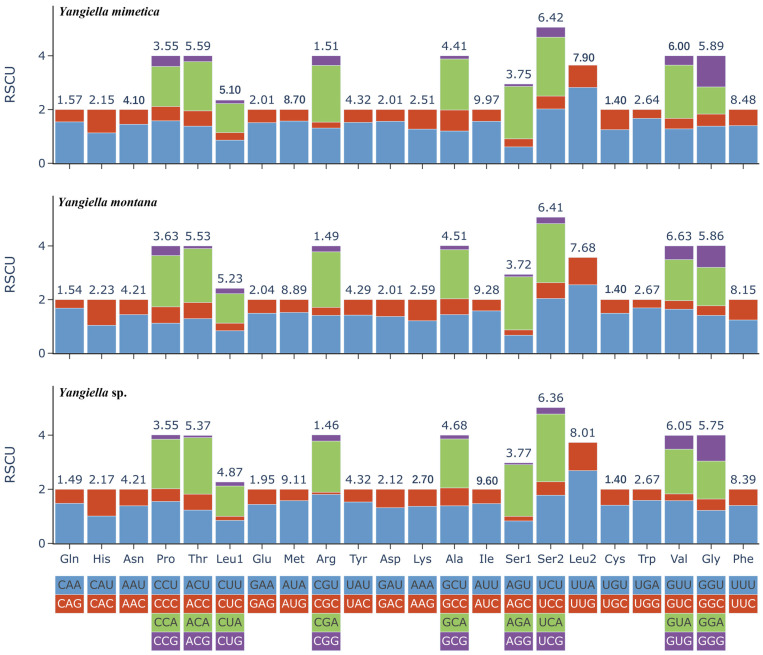
Relative synonymous codon usage (RSCU) in the mitogenomes of the three species. The numbers above the bar graph indicate the frequency of amino acids. The number of codons per amino acid varies from 2 to 4. The RSCU values are color-coded based on the codons below the amino acid labels.

**Figure 5 insects-15-00533-f005:**
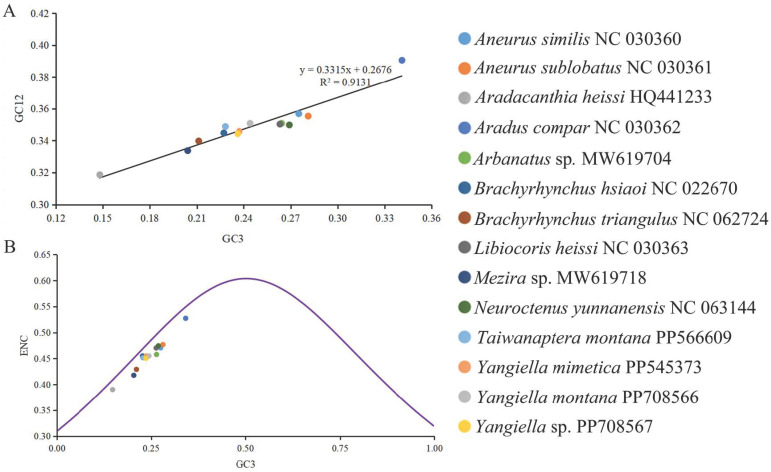
Codon preference analysis of 14 species in Aradidae. (**A**) The G + C content trend line and R value of codon 1 and 2 (GC12) and codon 3 (GC3) of mitochondrial protein-coding gene in Aradidae. (**B**) The scatter plot of the correlation between ENC value and GC3. The purple curve represents the expected functional relationship between ENC and GC3 under no selection pressure but only under abrupt pressure.

**Figure 6 insects-15-00533-f006:**
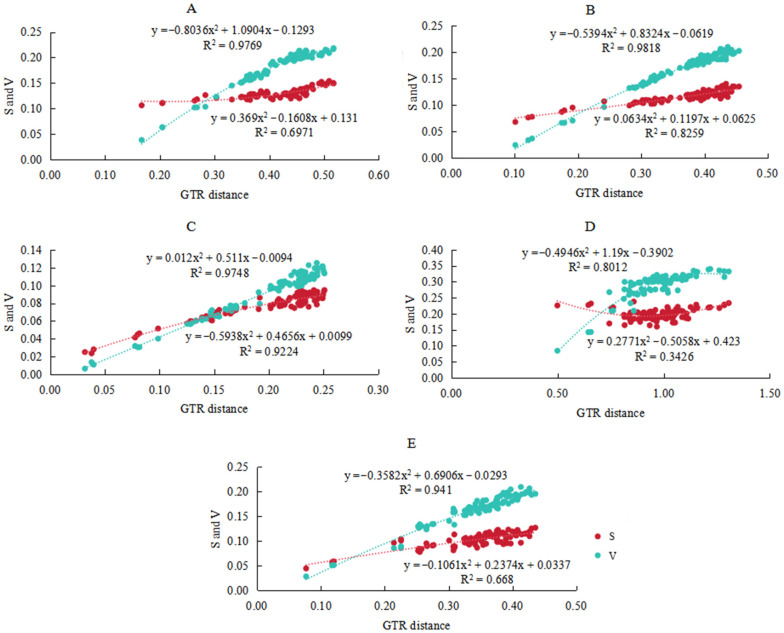
Substitution saturation plots of PCGs and RNA in the mt genomes of 14 species of Aradidae. “S” represents the transition rate (dark red), and “V” represents the transversion rate (dark green). (**A**) All sites of protein-coding genes. (**B**) The first site of protein-coding genes. (**C**) Protein-coding gene second site. (**D**) Protein-coding gene third site. (**E**) rRNA gene.

**Figure 7 insects-15-00533-f007:**
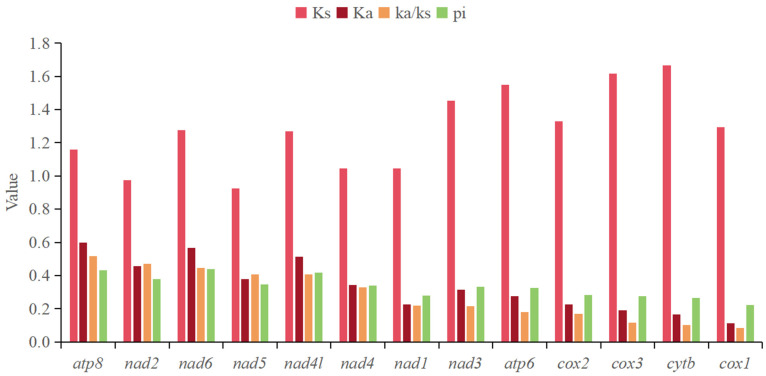
Nucleotide diversity and Ka/Ks values of PCGs in mitochondrial genomes of 14 species of Aradidae.

**Figure 8 insects-15-00533-f008:**
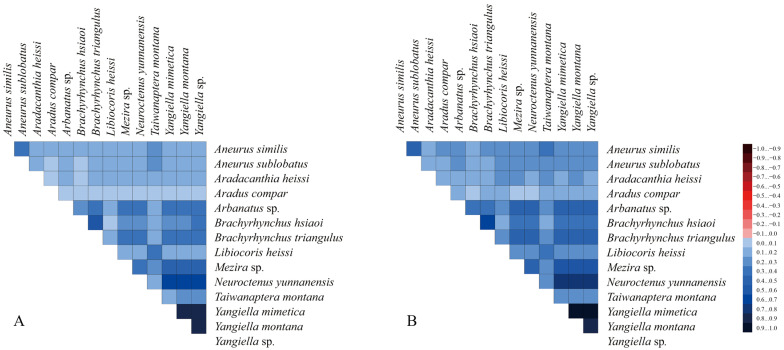
Heterogeneity of the sequence composition of the mitochondrial genomes in different datasets. The pairwise Aliscore values are indicated by colored squares. Darker colors indicate full random similarity, and lighter colors indicate the opposite. (**A**) PCGsRNA matrix. (**B**) PCGs12RNA matrix.

**Figure 9 insects-15-00533-f009:**
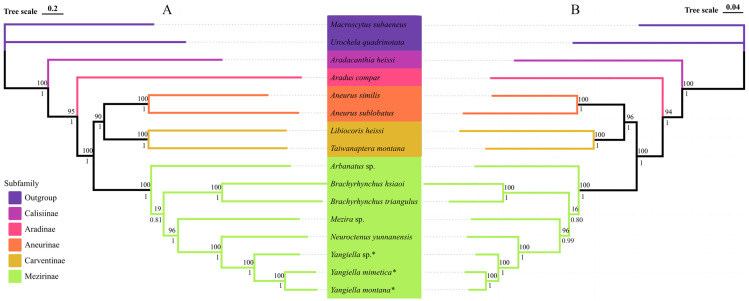
Phylogenetic tree of 16 species of Hemiptera inferred from the PCGsRNA and PCGs12RNA data matrix using maximum likelihood (ML) and Bayesian inference (BI). The ML tree has the same topology as the BI tree, and their support values are reported above and below the nodes, respectively. The species marked as * is the research object of this study. (**A**) Phylogenetic tree based on PCGsRNA data matrix. (**B**) Phylogenetic tree based on PCGs12RNA data matrix.

**Figure 10 insects-15-00533-f010:**
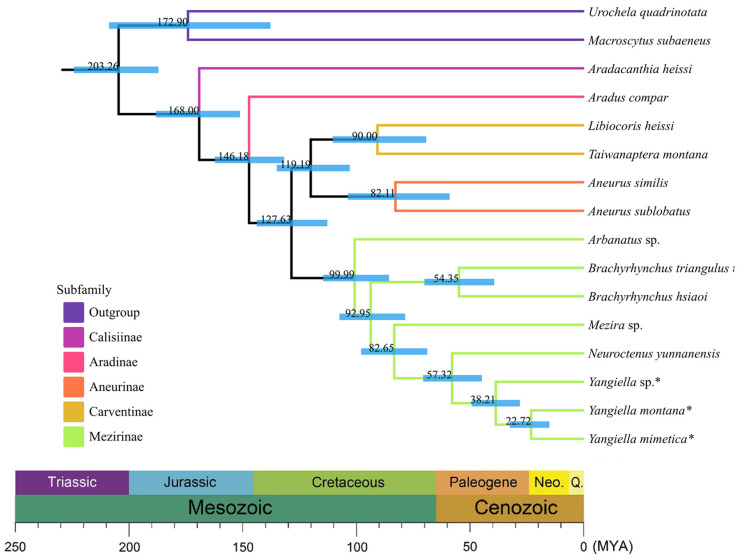
Chronogram showing Aradidae phylogeny and divergence time. The consensus tree shows the results of the divergence time estimation of the PCGsRNA dataset using three calibration points (Appendix A). The species marked as * is the research object of this study. A geological time scale is shown at the bottom.

**Table 1 insects-15-00533-t001:** Voucher information of the specimens used for mitochondrial genome sequencing.

Specimens	Date of Collection	Collection Site	Longitude (E)	Latitude (N)	GB Numbers
*Y. mimetica*	16 January 2011	Jinping, Yunnan	103.2383	22.9067	PP545373
*Y. montana*	31 October 2019	Yingjiang Tongbiguan, Yunnan	97.9364	24.7031	PP708566
*Yangiella* sp.	27 May 2023	Shuangbai Dutian Forest farm, Yunnan	101.4556	24.5447	PP708567

## Data Availability

The genomic data in this study are available under the NCBI accession numbers PP708566, PP545373, and PP708567.

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
