# Peer review of "Comparative Analysis of the Mitochondrial Genomes of Three Species of Yangiella (Hemiptera: Aradidae) and the Phylogenetic Implications of Aradidae"

_insects, 2024, doi:10.3390/insects15070533_

Round 1

Reviewer 1 Report

Comments and Suggestions for Authors

In the reviewed MS the authors analyzed 3 mitogenomes of mizerine insects. They give comparative data on the mitogenomes, performed a series analyses, and provided some general conclusions, based on their calculations. The MS has a lot methodological and stylistic flows. The dataset which the authors used for their analyses need a better explanation as well as methodologies used for reconstructing phylogenetic trees. The authors describe in detail the data from Tables, it is better to avoid this. The Figures need revisions. Some parts of the MS need to be written in a notably shorter way. Discussion and Conclusions need serious revision. Some additional comments are below.

8: Aradidae is a mysterious --- do you really need “misterious” here? Please, explain why do you use this word here

14: On this basis, --- this sounds logically inconsistent

15: phylogenetic analysis -à analyses

18: In this study,  --- redundant

20-21: may be “typical gene number” is better

24: gene rearrangement in Aradidae was observed, --- please specify which rearrangement was observed

27-28 Substitution saturation level analysis showed that the nucleic acid sequence of mitochondrial protein-coding genes in Aradidae did not reach saturation. --- please add something like: … suggesting that… or indicating that… and give your interpretation of this result.

29: At the same time --- redundant

30: which proved the monophyly of each subfamily within Aradidae --- please, specify the subfamilies

31: The divergence time was inferred, --- please, add “based on…” and briefly explain the origin of the calibration points which you used

57: 57 million years ago --- do you have ecological/climatic exaplnation for this?

39: Larger --- larger

40: Yangiella is a genus of Mezirinae[2]. --- please, give some brief information on this subfamily and explain why the genus Yangiella is interesting/important ?

45-48: obvious, redundant, may be removed

52-53: were any other genes than cox1 used for investigating phylogeny of this taxon or not? Please, be more precise

56-57: this sentence may need revision and references

58-59: please, say directly which advantages and limitations it has

63: and so on --- remove this

61-68: please, specify directly the goal of your study and based on this goal mention what exactly was done to achieve this goal

73: “to an environment” --- do you really need to say this?

74: head, chest and legs of each specimen --- please, explain this better. Do you mean that you have 3 genomes for each specimen?

75: please, specify the DNA concentration

78-79: please specify the coverage, the genome size and how many Gb for each sample were obtained

89: Neuroctenus yunnanensis --- please specify higher taxonomy of this species

95: it seems that you missed a verb in the first part of this sentence

97: Server online server (https://cgview.ca/) --- something is wrong with this reference, it leads to SPADES: https://github.com/ablab/spades

98-99: it is unclear why did you compare the mitogenomes with Drosophilla? Please, compare it with closest, close and distant taxa and perform gene order analysis

100-102: GB numbers could be given in the Table 1 and these 3 lines could be removed

109: Neutrality-polt -à plot

120-123: Please explain better in detail how the dataset was constructed. There are currently 23 mitogenomes of the targeted taxon in the GenBank, please use all of them and explain the choice of out-groups

148: and beautify --- redundant

154: The divergence time of the previous study was used as the calibration point --- this is unclear, please specify in detail how did you calibrate the tree, which calibration points were used, and what are the origins of these points?

156: The total algebra --- this needs rewording

161: MCC tree and beautify it. --- what is MCC? “Beautify” is redundant

172: with most of the flat bugs --- please specify taxa

173: Aradus comper, Aradacanthia heissi --- italic, check italic in the whole MS

Fig1. The figures of the three mitogenomes are suboptimal, it is hard to read the gene names

198: Other species of Aradidae --- please, specify the taxa

203: 69 bp ; the --- check the space between words

204: DHU arm --- please explain this abbreviation, because not all readers are familiar with it

Fig3: please add the explanation of different colors to the caption. You may refer to the last tRNA in the Figure. No need to show all tRNA here. You may leave only S1 and S2 in the Fig2 in the main text and give all other tRNAs in Supplement

Table 3 --- please move this Table to Supplement

219: First bullet; --- remove

230: Neutral plot analysis results (Figure 5) : --- the results of … are shown…

Section 3.3. --- this section needs rewriting

Fig.4.5 --- these figures need better explanation, e.g. like you did in lines 263-266 (R value, ECN, correlation and the mean of this correlation). Usually, some terms and the idea of the analysis is described in Material and Methods or directly prior to presenting results (like 263-266).

247: In this study, the substitution rates of mitochondrial protein-coding genes and rRNA genes were calculated (Figure 6). The results show that --- redundant

253: it does not reach saturation at all --- needs rewording

Section 3.4 --- this section needs rewriting. Please write it shorter, list the main results and give proves

267-273: this is a well-known fact, that cx1 and cyb are more conservative. Please, give references and compare with data from literature on this topic

Fig.6 and 7 --- 14 species ---please, explain which species do you mean here

Section 3.5 needs rewording

Section 3.6. analysis --- analyses

Fig. 8. Please revise the caption

298: Pentatomoidea --- you need to say something about this taxon in material and Method when you describe your dataset

Fig. 9,10 – please, revise the captions

Fig. 10 --- please provide the original time tree produced by the program, which you used. Please check in some molecular phylogenetic journals how time is usually presented

319 – repetition, redundant

Discussion needs rewriting in a shorter, more condensed form

Conclusion needs revisions. Please, report the main results of the study and underline their importnace

S6,S7,S8 – minimum\maximum --- please revise (start – stop/end)

S2,s3,s4,s5 – site --- please revise this word

Comments on the Quality of English Language

minor-moderate

Author Response

Comments 1: 8: Aradidae is a mysterious --- do you really need “misterious” here? Please, explain why do you use this word here.

Response 1: Thank you for pointing this out. Because Pentatomidae lives in a hidden place, it is not common in daily life, and there are few studies on the species of this family. So I used “misterious” to describe it.

Comments 2: 14: On this basis, --- this sounds logically inconsistent

Response 2: Thank you for your careful review. It is indeed illogical, I have rewritten this part. The revised sentence is located on line 14 of the revised manuscript.

Comments 3: 15: phylogenetic analysis -à analyses

Response 3 : Thank you for pointing this out. The error in this section was modified. The revised sentence is located on line 15 of the revised manuscript.

Comments 4: 18: In this study,  --- redundant

Response 4: I deleted this phrase

Comments 5: 20-21: may be “typical gene number” is better

Response 5: Thank you for your advice, I made the appropriate changes to make this sentence more fluent. The revised sentences are located in lines 19-21 of the revised manuscript.

Comments 6: 24: gene rearrangement in Aradidae was observed, --- please specify which rearrangement was observed

Response 6: Thanks for your suggestion, the sentence has been modified to ' the rearrangement of tRNA genes was observed in Aradidae '. The revised sentences are located in lines 19-21 of the revised manuscript.

Comments 7: 27-28 Substitution saturation level analysis showed that the nucleic acid sequence of mitochondrial protein-coding genes in Aradidae did not reach saturation. --- please add something like: … suggesting that… or indicating that… and give your interpretation of this result.

Response 7: Thank you for your suggestion, this sentence has been supplemented accordingly. This sentence is located in lines 19-21 of the revised manuscript.

Comments 8: 29: At the same time --- redundant

Response 8: Thank you for pointing this out. I deleted this short sentence

Comments 9: 30: which proved the monophyly of each subfamily within Aradidae --- please, specify the subfamilies.

Response 9: Thank you for pointing out the problem, I added the corresponding subfamily name. This sentence is located on lines 31-32 of the revised manuscript.

Comments 10: 31: The divergence time was inferred, --- please, add “based on…” and briefly explain the origin of the calibration points which you used.

Response 10: Thank you for your advice, I have made a corresponding supplement to this sentence. This sentence is located in lines 32-33 of the revised manuscript.

Comments 11: 57 million years ago --- do you have ecological/climatic exaplnation for this?

Response 11: There is no in-depth study on climate and ecology, and the divergence time is inferred based on fossils and previous studies.

Comments 12: 39: Larger --- larger

Response 12: I am very sorry for such a mistake, I have modified it. The word is located on line 41 of the revised manuscript.

Comments 13: Yangiella is a genus of Mezirinae[2]. --- please, give some brief information on this subfamily and explain why the genus Yangiella is interesting/important ?

Response 13: According to your suggestions, I made a corresponding supplement to this part. This part is located in lines 43-43 and 46-48 of the revised manuscript.

Comments 14: 45-48: obvious, redundant, may be removed

Response 14: I am very sorry, this paragraph is indeed redundant, I have deleted this paragraph according to your suggestion.

Comments 15: 52-53: were any other genes than cox1 used for investigating phylogeny of this taxon or not? Please, be more precise

Response 15: I ' m very sorry that my description is too vague. I have described this part in detail according to your suggestion. This part is located in line 56-59 of the revised manuscript.

Comments 16: 56-57: this sentence may need revision and references

Response 16: Thank you for asking this question, I rewrote this part and made a corresponding supplement. This part is located in line 60-62 of the revised manuscript.

Comments 17: 58-59: please, say directly which advantages and limitations it has

Response 17: Thank you for your suggestion, I added the advantages and disadvantages according to your suggestion. This part is located in line 62-66 of the revised manuscript.

Comments 18: 63: and so on --- remove this

Response 18: Thank you for pointing this out. This phrase was deleted.

Comments 19: 61-68: please, specify directly the goal of your study and based on this goal mention what exactly was done to achieve this goal

Response 19: Thank you very much for your suggestion. This passage is indeed a little vague. I have rewritten this part according to your suggestion. This part is located in line 67-75 of the revised manuscript.

Comments 20: 73: “to an environment” --- do you really need to say this?

Response 20: Thank you for pointing out this problem. The sentence is more refined after the phrase is deleted.

Comments 21: 74: head, chest and legs of each specimen --- please, explain this better. Do you mean that you have 3 genomes for each specimen?

Response 21: I ' m sorry that this sentence is ambiguous, so I rewrite it. Because the abdomen of the specimen may be contaminated by other biological DNA, the genomic DNA of the head, chest and foot is extracted, and not each specimen has three genomes. This part is located in line 82-84 of the revised manuscript.

Comments 22: 75: please, specify the DNA concentration

Response 22: Thank you for pointing out this problem, which has been added in the sentence ( DNA concentration ≥ 20ng / ul, total ≥ 500ng ). This part is located in line 85-86 of the revised manuscript.

Comments 23: 78-79: please specify the coverage, the genome size and how many Gb for each sample were obtained

Response 23: The clean data size of a single species is only 4-5G, which is supplemented in the 2.2 section. Due to the mitochondrial genome we studied, the sequencing depth cannot accurately estimate the size of the whole genome, and can only ensure the correct assembly of mitochondria. In this regard, we can provide a coverage depth figure of the corresponding mitochondrial genome. This part is located in line 94 of the revised manuscript.

Comments 24: 89: Neuroctenus yunnanensis --- please specify higher taxonomy of this species.

Response 24: Thank you for your suggestion, I supplemented the species classification data. This part is located in line 97-98 of the revised manuscript.

Comments 25: 95: it seems that you missed a verb in the first part of this sentence

Response 25: Thank you for your advice, but I examined the sentence carefully and found no lack of verbs. This part is located in line 97-98 of the revised manuscript.

Comments 26: 97: Server online server (https://cgview.ca/) --- something is wrong with this reference, it leads to SPADES: https://github.com/ablab/spades

Response 26: I 'm sorry to insert the wrong hyperlink, I have modified it. This hyperlink is located on line 101 of the revised manuscript.

Comments 27: 98-99: it is unclear why did you compare the mitogenomes with Drosophilla? Please, compare it with closest, close and distant taxa and perform gene order analysis

Response 27: Thank you for pointing this out. Because previous studies have found that the mitochondrial structure of Drosophilla is the arrangement pattern of insect ancestors, Drosophilla mitochondria were used for comparison.

Comments 28: 100-102: GB numbers could be given in the Table 1 and these 3 lines could be removed

Response 28: Thank you for pointing this out. I have moved these three login numbers into Table 1.

Comments 29: 109: Neutrality-polt -à plot

Response 29: Thanks for your suggestion, I delete unnecessary dashes. This part is located on line 118 of the revised manuscript.

Comments 30: 120-123: Please explain better in detail how the dataset was constructed. There are currently 23 mitogenomes of the targeted taxon in the GenBank, please use all of them and explain the choice of out-groups

Response 30: Thank you for pointing this out. I have modified and supplemented this part. Explain your problem below. Not all the mitochondrial genomes of this group on NCBI were used because our team was skeptical about the accuracy of the mitochondrial genomes of some species on NCBI.In subsequent studies, we will re-sequence and upload those species that are in doubt. Finally, the reason why the two species are selected as outgroups is that the suborders of these two species can use fossil correction points.

Comments 31: 148: and beautify --- redundant

Response 31: According to your suggestion, I deleted these unnecessary words.

Comments 32: 154: The divergence time of the previous study was used as the calibration point --- this is unclear, please specify in detail how did you calibrate the tree, which calibration points were used, and what are the origins of these points?

Response 32: I am very sorry, this is my writing negligence. I have supplemented these data in the supplementary table S6.

Comments 33: 156: The total algebra --- this needs rewording

Response 33: I am very sorry for this error. I have changed “ algebra “ to “ generations “. This part is located on line 166 of the revised manuscript.

Comments 34: 161: MCC tree and beautify it. --- what is MCC? “Beautify” is redundant

Response 34: Thank you for pointing this out. The MCC tree is the maximum clade credibility tree, and I remove unnecessary words.

Comments 35: 172: with most of the flat bugs --- please specify taxa

Response 35: Thank you for your suggestion, I have supplemented the classification information in detail. And move this part to line 340-341 of the discussion section.

Comments 36: 173: Aradus comper, Aradacanthia heissi --- italic, check italic in the whole MS

Response 36: This is a gross omission, thank you for pointing out. I re-examined the whole article.

Comments 37: Fig1. The figures of the three mitogenomes are suboptimal, it is hard to read the gene names

Response 37: Thank you for pointing this out. The gene name in Figure 1 is re-described to make it easy to understand.

Comments 38: 198: Other species of Aradidae --- please, specify the taxa

Response 38: Thank you very much for your suggestion, the description is too vague, I have revised it to Aneurinae, Carventinae and Mezirinae. This part is located on line 206 of the revised manuscript.

Comments 39: 203: 69 bp ; the --- check the space between words

Response 39: Thank you very much for pointing this out, and I ' ll recheck the full text to see if I ' ve added spaces in the appropriate places.

Comments 40: 204: DHU arm --- please explain this abbreviation, because not all readers are familiar with it

Response 40: Thank you for pointing this out. The corresponding full name was added after " DHU arm ".This part is located on line 211 of the revised manuscript.

Comments 41: Fig3: please add the explanation of different colors to the caption. You may refer to the last tRNA in the Figure. No need to show all tRNA here. You may leave only S1 and S2 in the Fig2 in the main text and give all other tRNAs in Supplement

Response 41: Thank you very much for pointing out the problem, I explained the structure of different colors according to your suggestion in the title of Figure 3. However, I think it may be more beautiful to put all tRNAs on a picture, and the characteristics of the tRNA secondary structure of the species can be seen more intuitively.

Comments 42: Table 3 --- please move this Table to Supplement

Response 42: There is no Table 3 in the text, but I guess you mean to move Table 2 to the appendix, I made the corresponding changes.

Comments 43: 219: First bullet; --- remove

Response 43: Thank you for pointing this out. I deleted the phrase.

Comments 44: 230: Neutral plot analysis results (Figure 5) : --- the results of … are shown…

Response 44: According to your suggestion, I rewrite this sentence. This part is located on line 241 of the revised manuscript.

Comments 45: Section 3.3. --- this section needs rewriting

Response 45: Thank you for pointing out this part of the problem. I rewrote Section 3.3 and deleted some unnecessary descriptions to make it more refined. This part is located on line 241-246 of the revised manuscript.

Comments 46: Fig.4.5 --- these figures need better explanation, e.g. like you did in lines 263-266 (R value, ECN, correlation and the mean of this correlation). Usually, some terms and the idea of the analysis is described in Material and Methods or directly prior to presenting results (like 263-266)

Response 46: Thank you for pointing out this problem. I explain the numbers in Figure 4 and Figure 5 in the materials and methods and conclusions. And the explanation is given in the title of Figure 4.

Comments 47: 247: In this study, the substitution rates of mitochondrial protein-coding genes and rRNA genes were calculated (Figure 6). The results show that --- redundant

Response 47: Thank you for your suggestion, this sentence is indeed a bit redundant, I have deleted it.

Comments 48: 253: it does not reach saturation at all --- needs rewording

Response 48: On your suggestion, I have rewritten the sentence. This part is located on line 259 of the revised manuscript.

Comments 49: Section 3.4 --- this section needs rewriting. Please write it shorter, list the main results and give proves

Response 49: Following your suggestion, I have rewritten the 3.4 part to make it more refined.

Comments 50: 267-273: this is a well-known fact, that cx1 and cyb are more conservative. Please, give references and compare with data from literature on this topic

Response 50: Thank you for your advice, I added references and made a comparative analysis. But I put this part into the discussion. This part is located on line 259 of the revised manuscript.

Comments 51: Fig.6 and 7 --- 14 species ---please, explain which species do you mean here

Response 51: The classification data of the 14 species in Fig.6 and Fig.7 were supplemented.

Comments 52: Section 3.5 needs rewording

Response 52: Thank you for your comments. There are too many unnecessary sentences in the 3.5 section. I rewrote this section at your suggestion.

Comments 53: Section 3.6. analysis --- analyses

Response 53: The word errors were modified.

Comments 54: Fig. 8. Please revise the caption

Response 54: I modified the caption to make it more accurate.

Comments 55: 298: Pentatomoidea --- you need to say something about this taxon in material and Method when you describe your dataset.

Response 55: I 'm very sorry, this is a negligence of mine. I have supplemented the relevant classification information in Table S1 according to your suggestions.

Comments 56: Fig. 9,10 – please, revise the captions

Response 56: According to your suggestion, I modified the title of these two figure.

Comments 57: Fig. 10 --- please provide the original time tree produced by the program, which you used. Please check in some molecular phylogenetic journals how time is usually presented

Response 57: Thank you very much for your suggestion. Figure 10 is indeed a little rough. According to your suggestion, I made this Figure again.

Comments 58: 319 – repetition, redundant

Response 58: I deleted this sentence and rewritten the part.

Comments 59: Discussion needs rewriting in a shorter, more condensed form 

Response 59: Thank you very much for your suggestion, I have rewritten the discussion section to make it more refined.

Comments 60: Conclusion needs revisions. Please, report the main results of the study and underline their importnace

Response 60: My conclusion part is somewhat vague before writing, according to your suggestion I modified this part.

Comments 61: S6,S7,S8 – minimum\maximum --- please revise (start – stop/end)

Response 61: Thank you very much for your suggestion, I have modified it.

Comments 62: S2,s3,s4,s5 – site --- please revise this word

Response 62: I modified this word to “ Number of sites “.

Reviewer 2 Report

Comments and Suggestions for Authors

The reviewed article contains an in-depth analysis of the phylogenetic similarity of three species of bedbugs from the genus Yangiella. The authors have a high level of knowledge of modern methods of genetic analysis. This article is a significant achievement in the phytogeny of bugs. It will be of interest not only to geneticists, but also to general entomologists.

Comments on individual elements of the article.

1. Line 39-44: readers will be interested to learn about the ecology of bedbugs of this family and the trophic relationships of species of the genus Yangiella.

2. In accordance with the International Code of Zoological Nomenclature, the first mention of the Latin name of a species in the text and in Table 1 must be accompanied by the name of the author and the year. This applies to all animal species mentioned in the article, not just those from the genus Yangiella.

3. The Results section should contain only your own results, without comparison and interpretation, without elements of the methodology. References to literature are not allowed in the Results section.

4. All figures in Table 2 in the “Nucleotides Composition (%)” columns must be rounded to the nearest tenth. All figures in the last two columns of this table should be rounded to the nearest thousand. Carelessness is unacceptable.

5. There is often no space between words, as well as before references to literature sources. Carelessness is unacceptable.

6. In the names of the axes on the graphs, you should always indicate “Name of characteristic, comma, unit of measurement.” For example, in Figure 5 and other figures.

7. All numbers indicated above the bars in Figure 4 must be rounded to the nearest hundredth.

8. In Figure 5 (and other figures), do not use bold fonts.

9. On the abscissa axis of Figure 5a, the range of values ​​should be selected from 0.12 to 0.36 in increments of 0.03. On the ordinate - from 0.30 to 0.42 in increments of 0.02.

10. On the abscissa axis of Figure 5, round all numbers to hundredths. In other figures, the rounding of all numbers on the axis should also be the same.

11. In Figure 6, the horizontal size of the image needs to be increased by 40-50%, and vertically by 100-120%. The size of letters and numbers in all figures should not be larger or smaller than the size of letters and numbers in the text of the article. Do not use bold font in pictures.

12. In Figure 7, discrete groups on the x-axis cannot be connected by a graph. This data should also be displayed as histogram bars (possibly in a different coordinate system). In the title of the figure and on the axes you need to indicate the unit of measurement (bits?). Genes on the x-axis should be italicized.

13. In Figures 8 and 10, after “sp” there should be a dot everywhere. The legend is very small, no one will read it. The size of all letters and numbers in all figures must be equal to the size of the letters in the article.

14. Line 276, error.

15. The title of Figure 8 is methodologically correct. First comes the general title of the picture, and then an explanation of the individual parts of the picture. The same cannot be said about the title of Figure 5.

16. There is no need for a space between initials in literature.

17. There is no need to write all words with a capital letter, for example line 502, 515, 528 and others.

18. Genes should be italicized, for example line 550.

Author Response

Comments 1: Line 39-44: readers will be interested to learn about the ecology of bedbugs of this family and the trophic relationships of species of the genus Yangiella.

Response 1: I made some supplements to this part, because there are not many studies on the ecology of the stink bugs in the family and the trophic relationship of the genus Yangiella, all of which can not be explained in detail. The modified part is located in lines 45-48 of the revised manuscript.

Comments 2: In accordance with the International Code of Zoological Nomenclature, the first mention of the Latin name of a species in the text and in Table 1 must be accompanied by the name of the author and the year. This applies to all animal species mentioned in the article, not just those from the genus Yangiella.

Response 2: Thank you for pointing out this problem, which is my negligence in writing. I have supplemented the author and year after the Latin name of the species that first appeared in the text as required, which has been highlighted in the text.

Comments 3: The Results section should contain only your own results, without comparison and interpretation, without elements of the methodology. References to literature are not allowed in the Results section.

Response 3: Thank you very much for your question, which is my negligence in writing. At your suggestion I have removed all references to the results section, describing only the results of this study.

Comments 4: All figures in Table 2 in the “Nucleotides Composition (%)” columns must be rounded to the nearest tenth. All figures in the last two columns of this table should be rounded to the nearest thousand. Carelessness is unacceptable.

Response 4: I am very sorry, this is my writing negligence, did not carefully check the form. According to your suggestion, I rechecked all the numbers of the table to ensure a uniform number of digits. 

Comments 5: There is often no space between words, as well as before references to literature sources. Carelessness is unacceptable.

Response 5: Thank you for pointing this out. I am very sorry for such a problem. It is a serious negligence that I did not check carefully before submitting it. In accordance with your recommendations, I re-examined the full text to prevent the recurrence of the same problem.

Comments 6: In the names of the axes on the graphs, you should always indicate “Name of characteristic, comma, unit of measurement.” For example, in Figure 5 and other figures.

Response 6: Thank you for pointing out this problem, I re-examined Figure in the text and supplemented them accordingly.

Comments 7: All numbers indicated above the bars in Figure 4 must be rounded to the nearest hundredth.

Response 7: Thank you very much for your reminder, I have modified the numbers on Figure 4.

Comments 8: In Figure 5 (and other figures), do not use bold fonts.

Response 8: This is an oversight in my writing. I checked all the figure according to your suggestion.

Comments 9: On the abscissa axis of Figure 5a, the range of values ​​should be selected from 0.12 to 0.36 in increments of 0.03. On the ordinate - from 0.30 to 0.42 in increments of 0.02. 

Response 9: After changing according to your suggestions, Figure 5 is more beautiful, thank you very much.

Comments 10: On the abscissa axis of Figure 5, round all numbers to hundredths. In other figures, the rounding of all numbers on the axis should also be the same.

Response 10: I am very sorry, this is my negligence, I have put the whole article picture of the inspection, to ensure that this problem will not happen again.

Comments 11: In Figure 6, the horizontal size of the image needs to be increased by 40-50%, and vertically by 100-120%. The size of letters and numbers in all figures should not be larger or smaller than the size of letters and numbers in the text of the article. Do not use bold font in pictures.

Response 11: Thank you very much for your suggestion, I have modified the size of the font as required and canceled all bold characters in the figure.

Comments 12: In Figure 7, discrete groups on the x-axis cannot be connected by a graph. This data should also be displayed as histogram bars (possibly in a different coordinate system). In the title of the figure and on the axes you need to indicate the unit of measurement (bits?). Genes on the x-axis should be italicized.

Response 12: According to your suggestion, all the data are shown as histograms, and the coordinate axes are unified. The unit of these data can only be represented by ' value ', and the gene name is also changed to italic.

Comments 13: In Figures 8 and 10, after “sp” there should be a dot everywhere. The legend is very small, no one will read it. The size of all letters and numbers in all figures must be equal to the size of the letters in the article.

Response 13: According to your suggestion, I have checked all the ' sp ' followed by whether the point has been added, while ensuring that the word in the figure is equal to the size of the text.

Comments 14: Line 276, error.

Response 14: Thank you very much for your reminder, I rewrote this sentence.

Comments 15: The title of Figure 8 is methodologically correct. First comes the general title of the picture, and then an explanation of the individual parts of the picture. The same cannot be said about the title of Figure 5.

Response 15: Thank you very much for your reminder, this is a serious negligence. I supplemented the title of Figure 5 according to the suggestion.

Comments 16: There is no need for a space between initials in literature.

Response 16: Thank you for pointing out this problem. I have re-examined the spaces in the initials of literature.

Comments 17: There is no need to write all words with a capital letter, for example line 502, 515, 528 and others.

Response 17: Thank you for pointing this out. Based on your comments, I checked these references. Those uppercase words are the names of some software, and they are all uppercase in the title of the original text, so I haven 't changed them.

Comments 18: Genes should be italicized, for example line 550.

Response 18: Thank you for your reminder, this is my negligence, very sorry. I re-examined the full text to prevent the same problem from happening again.

Reviewer 3 Report

Comments and Suggestions for Authors

Dear Authors,

thank you for very interesting and presisely done work.

The presented work is devoted to the study of mitochondrial genomes of three species of Yangiella, the hemipterans of the family Aradidae. Despite the great work done, the data obtained characterize only the metagenome found in three species, which can be used in further comparisons of the mitogenome, examining its changes. Of course, it is impossible to judge the evolution of families on the basis of just one character, even if revealed by genetic research methods, and, of course, it is impossible to build a phylogeny of families and distinguish taxa on this data alone. However, the authors understand this and carefully draw conclusions based on the data obtained. I would like to note one of the significant positive aspects in the work - the attentive attitude of the authors to taxonomic acts. Despite the identification of a new species, a new position in a number of taxa, the work does not provide either unsubstantiated names or new combinations, which could contradict the Code of Zoological Nomenclature.

I would like to ask a few questions to clarify the authors’ position on the topic of the work.

1. You write: “In summary, this study revealed 32 the mitochondrial characteristics of Yangiella, enriched the mitochondrial genetic information of 33 Aradidae species, and provided a reference for the subsequent taxonomy and genetics of Aradidae.” What special basis for subsequent taxonomy does the identified mitogen of 3 species provide that its data cover the importance of morphological and paleontological data? Without the data from that study, would the phylogeny be impossible to analyze?

2. The sentence sounds rather strange: “The reconstructed phylogenetic relationships strongly support the monophyly of the subfamilies and Yangiella of Aradidae in China.” Do you really think that monophyly can occur in species living in separate countries?

3. You talk about a new species of Yangiella, but do not characterize it, which is absolutely true, but still, where is it planned to publish a description of this species, and are there clear morphological differences between this new species and the two previously known?

The work leaves a good impression of the authors’ concern for the prospects for the development of the natural phylogeny. It is strongly recommended not to overestimate the importance of the mitogenome for phylogenetic reconstructions and to be based, nevertheless, on morphological and paleontological data, including complexes of characters that evolved in parallel with the morphoadaptation of species in changing ecosystems, and not only on mitogenome data. The author wishes to pay attention to this in the future.

Author Response

Comments 1: You write: “In summary, this study revealed 32 the mitochondrial characteristics of Yangiella, enriched the mitochondrial genetic information of 33 Aradidae species, and provided a reference for the subsequent taxonomy and genetics of Aradidae.” What special basis for subsequent taxonomy does the identified mitogen of 3 species provide that its data cover the importance of morphological and paleontological data? Without the data from that study, would the phylogeny be impossible to analyze?

Response 1:The three mitochondrial genomes can provide a partial molecular basis for subsequent taxonomic studies. However, this data cannot cover the importance of morphological and paleontological data. These data are complementary and need to verify each other to make the corresponding analysis. Without the data of this study, phylogenetic analysis will lack important molecular evidence..

Comments 2: The sentence sounds rather strange: “The reconstructed phylogenetic relationships strongly support the monophyly of the subfamilies and Yangiella of Aradidae in China.” Do you really think that monophyly can occur in species living in separate countries?

Response 2: To answer your question first, I don 't think monophyly will appear in species living in separate countries. Secondly, thank you very much for your reminder, because my insect specimens are from China, so when I write this sentence, I habitually add the description of ' in China ', but this is not suitable for describing monophyletic sentences.

Comments 3: You talk about a new species of Yangiella, but do not characterize it, which is absolutely true, but still, where is it planned to publish a description of this species, and are there clear morphological differences between this new species and the two previously known?

Response 3: Very sorry, this is a problem that needs to be solved, because it is impossible to determine the new species by only one mitochondria, and it needs to be confirmed together with morphological features. We originally planned to publish the description of the new species in Zootaxa, but due to some other problems, the new species has not been published yet. However, I am very sure that this new species has obvious morphological differences from the two known species ( such as antennae and pronotum ).

Round 2

Reviewer 1 Report

Comments and Suggestions for Authors

After revision, the MS looks better, however, probably the revision was made too fast and because of this the MS still needs further corrections. The MS needs moderate linguistic corrections including stylistic corrections. All figures and captions should be carefully revised. The gene order analysis needs revision. Some additional remarks are below.

8: “Aradidae is a mysterious species” --- Aradidae is a family. Remove “misterious”, this is a scientific paper, not mystic literature

24: rearrangement within Aradidae or in comparison with out-groups?

29: saturation,suggesting --- add a space

31: please, use more neutral verb instead of “proved”, because the dataset is small and any cladistics analysis only provides evidence in terms of probability

34-36 – redundant

43: world [2].Yangiella --- add a space, check through all MS

46-48: please, rewrite this sentence using scientific style

58: Marchal et al used --- please add a brief sentence indicating the main result of that study, e.g. which large lineages of aradids were found.

61-66: redundant or this may be written 2-3 times shorter

67-75: this paragraph needs revision

82: “the head, chest and legs of the specimen were separated” --- please, explain why did you do that and how did you use the DNA extracted from different parts of the specimen.

130-134: please provide the same explanation for the dataset, which you used in your “Answer to reviewer” file and say directly why you excluded some GB mitogenomes from your analyses.

182-186: the gene order analysis has been performed incorrectly and described suboptimally. Drosophila is too distant to conclude on gene order in Aradidae. Please, use close out-groups for comparison. Alternatively, just avoid speculations on the gene order rearrangements.

Figure 2. Please, indicate family and order for all insect species

Fig. 3. It is enough to give only S1 and S2. This would help to make them larger that now. It is hard to see the tRNA structures when all of them are shown. Please, move all other structures instead of S1 and S2 to Supplement

Fig. 4. Please, say in the caption what different colors mean

Fig. 6 please, add brackets E. rRNA gene.

Fig.6 please use more contrast colors, red+orange provide poor contrast. Please, explain colors in caption

Conclusions: please do not retell your MS but give distinct conclusions (logical consequences) based on the obtained results

Comments on the Quality of English Language

minor

Author Response

Comments 1: 8: “Aradidae is a mysterious species” --- Aradidae is a family. Remove “misterious”, this is a scientific paper, not mystic literature

Response 1: Thank you for your suggestion that words such as “misterious” are not suitable to appear in a scientific paper. I have deleted them and corrected the errors in the description of Aradidae. This section is located on lines 8-9 of the revised manuscript.

Comments 2: 24: rearrangement within Aradidae or in comparison with out-groups?

Response 2: I am very sorry, the description here is too vague, I have been added. Rewritten as " tRNA gene rearrangements were observed among the three subfamilies of Aradidae ( Mezirinae, Calisiinae, Aradinae ) ". This section is located on lines 24-25 of the revised manuscript.

Comments 3: 29: saturation,suggesting --- add a space

Response 3: Thank you very much for your reminder, this is my negligence, did not notice a space is missing here. This section is located on lines 30 of the revised manuscript.

Comments 4: 31: please, use more neutral verb instead of “proved”, because the dataset is small and any cladistics analysis only provides evidence in terms of probability

Response 4: Thank you for pointing this out. There is something really wrong with the word " proved", and I have changed it to the more neutral word " supported ". The word is located in lines 32 of the revised manuscript.

Comments 5: 34-36 – redundant

Response 5: I am very sorry, this part of the paragraph is not necessary to write. I have deleted it according to your suggestion.

Comments 6: 43: world [2].Yangiella --- add a space, check through all MS

Response 6: Thank you for pointing out this problem. At the same time, I am very sorry that such negligence has occurred again. I will check the full text again. This section is located on lines 41 of the revised manuscript.

Comments 7: 46-48: please, rewrite this sentence using scientific style

Response 7: I am very sorry that this sentence should not appear in the scientific paper, I have rewritten it. This section is located on lines 44-48 of the revised manuscript.

Comments 8: 58: Marchal et al used --- please add a brief sentence indicating the main result of that study, e.g. which large lineages of aradids were found.

Response 8: Thank you for pointing out this problem, I supplemented this part. This section is located on lines 44-48 of the revised manuscript. This section is located on lines 56-59 of the revised manuscript.

Comments 9: 61-66: redundant or this may be written 2-3 times shorter

Response 9: Thank you for your suggestion. This part is indeed a little redundant. I have abbreviated this part. This section is located on lines61-63 of the revised manuscript.

Comments 10: 67-75: this paragraph needs revision

Response 10: Thank you for your suggestion, this part of the paragraph is not refined enough, according to the suggestion I rewrite this part. This section is located on lines 64-72 of the revised manuscript.

Comments 11: 82: “the head, chest and legs of the specimen were separated” --- please, explain why did you do that and how did you use the DNA extracted from different parts of the specimen.

Response 11: Thank you for pointing out this problem, I rewrite this part, but the specific steps are too cumbersome I did not describe too much. This section is located on lines 70-82 of the revised manuscript.

Comments 12: 130-134: please provide the same explanation for the dataset, which you used in your “Answer to reviewer” file and say directly why you excluded some GB mitogenomes from your analyses.

Response 12: Thank you for your reminder, the lack of some species data is really confusing. According to the suggestion, I move the explanation in the " Answer to reviewer " file to the footer of Table S1 for explanation.

Comments 13: 182-186: the gene order analysis has been performed incorrectly and described suboptimally. Drosophila is too distant to conclude on gene order in Aradidae. Please, use close out-groups for comparison. Alternatively, just avoid speculations on the gene order rearrangements.

Response 13: Thank you very much for pointing this out, Drosophila is too genetically distant for comparisons to be meaningful. According to your opinion, I chose the insects belonging to Heteroptera for comparison. These revisions are located in lines 104-106, 179-180 and 343-345 of the revised manuscript.

Comments 14: Figure 2. Please, indicate family and order for all insect species

Response 14: Thank you very much for your suggestion, I added specific species in the title of Figure 2.

Comments 15: Fig. 3. It is enough to give only S1 and S2. This would help to make them larger that now. It is hard to see the tRNA structures when all of them are shown. Please, move all other structures instead of S1 and S2 to Supplement

Response 15: Thank you very much for pointing this out, as you suggested I kept only S1 and S2 in Figure 3 and put the rest in the supplementary material. This makes the secondary structure of tRNAs in the main text clearer, thank you very much.

Comments 16: Fig. 4. Please, say in the caption what different colors mean

Response 16: Thank you for pointing this out, as per your request I explained the meaning of the colors in Figure 4.

Comments 17: Fig. 6 please, add brackets E. rRNA gene.

Response 17: Very sorry for this oversight. I have added accordingly

Comments 18: Fig.6 please use more contrast colors, red+orange provide poor contrast. Please, explain colors in caption

Response 18: I'm very sorry, it was a mistake in my choice of colors. As you suggested I chose a contrasting dark green and dark red.

Comments 19: Conclusions: please do not retell your MS but give distinct conclusions (logical consequences) based on the obtained results

Response 19: Thank you very much for pointing out this part of the problem, it was very uncalled for to repeat too much content in my conclusion. I have rewritten this section as you suggested.